# Establishing In-House Cutoffs of CSF Alzheimer’s Disease Biomarkers for the AT(N) Stratification of the Alzheimer Center Barcelona Cohort

**DOI:** 10.3390/ijms23136891

**Published:** 2022-06-21

**Authors:** Adelina Orellana, Pablo García-González, Sergi Valero, Laura Montrreal, Itziar de Rojas, Isabel Hernández, Maitee Rosende-Roca, Liliana Vargas, Juan Pablo Tartari, Ester Esteban-De Antonio, Urszula Bojaryn, Leire Narvaiza, Emilio Alarcón-Martín, Montserrat Alegret, Daniel Alcolea, Alberto Lleó, Lluís Tárraga, Vanesa Pytel, Amanda Cano, Marta Marquié, Mercè Boada, Agustín Ruiz

**Affiliations:** 1Ace Alzheimer Center Barcelona, Universitat Internacional de Catalunya (UIC), 08029 Barcelona, Spain; aorellana@fundacioace.org (A.O.); pgarcia@fundacioace.org (P.G.-G.); svalero@fundacioace.org (S.V.); lmontrreal@fundacioace.org (L.M.); iderojas@fundacioace.org (I.d.R.); ihernandez@fundacioace.org (I.H.); mrosende@fundacioace.org (M.R.-R.); lvargas@fundacioace.org (L.V.); jptartari@fundacioace.org (J.P.T.); eesteban@fundacioace.org (E.E.-D.A.); ubojaryn@fundacioace.org (U.B.); lnarvaiza@fundacioace.org (L.N.); ealarcon@fundacioace.org (E.A.-M.); malegret@fundacioace.org (M.A.); ltarraga@fundacioace.org (L.T.); vpytel@fundacioace.org (V.P.); acano@fundacioace.org (A.C.); mmarquie@fundacioace.org (M.M.); mboada@fundacioace.org (M.B.); 2Biomedical Research Networking Centre in Neurodegenerative Diseases (CIBERNED), 28031 Madrid, Spain; dalcolea@santpau.cat (D.A.); alleo@santpau.cat (A.L.); 3Sant Pau Memory Unit, Department of Neurology, Hospital de la Santa Creu i Sant Pau, Biomedical Research Institute Sant Pau, Universitat Autònoma de Barcelona, 08029 Barcelona, Spain

**Keywords:** cerebrospinal fluid, Alzheimer’s disease, chemiluminescent enzyme immunoassay, Lumipulse, MCI

## Abstract

Background: Clinical diagnosis of Alzheimer’s disease (AD) increasingly incorporates CSF biomarkers. However, due to the intrinsic variability of the immunodetection techniques used to measure these biomarkers, establishing in-house cutoffs defining the positivity/negativity of CSF biomarkers is recommended. However, the cutoffs currently published are usually reported by using cross-sectional datasets, not providing evidence about its intrinsic prognostic value when applied to real-world memory clinic cases. Methods: We quantified CSF Aβ1-42, Aβ1-40, t-Tau, and p181Tau with standard INNOTEST^®^ ELISA and Lumipulse G^®^ chemiluminescence enzyme immunoassay (CLEIA) performed on the automated Lumipulse G600II. Determination of cutoffs included patients clinically diagnosed with probable Alzheimer’s disease (AD, n = 37) and subjective cognitive decline subjects (SCD, n = 45), cognitively stable for 3 years and with no evidence of brain amyloidosis in 18F-Florbetaben-labeled positron emission tomography (FBB-PET). To compare both methods, a subset of samples for Aβ1-42 (n = 519), t-Tau (n = 399), p181Tau (n = 77), and Aβ1-40 (n = 44) was analyzed. Kappa agreement of single biomarkers and Aβ1-42/Aβ1-40 was evaluated in an independent group of mild cognitive impairment (MCI) and dementia patients (n = 68). Next, established cutoffs were applied to a large real-world cohort of MCI subjects with follow-up data available (n = 647). Results: Cutoff values of Aβ1-42 and t-Tau were higher for CLEIA than for ELISA and similar for p181Tau. Spearman coefficients ranged between 0.81 for Aβ1-40 and 0.96 for p181TAU. Passing–Bablok analysis showed a systematic and proportional difference for all biomarkers but only systematic for Aβ1-40. Bland–Altman analysis showed an average difference between methods in favor of CLEIA. Kappa agreement for single biomarkers was good but lower for the Aβ1-42/Aβ1-40 ratio. Using the calculated cutoffs, we were able to stratify MCI subjects into four AT(N) categories. Kaplan–Meier analyses of AT(N) categories demonstrated gradual and differential dementia conversion rates (*p* = 9.815^−27^). Multivariate Cox proportional hazard models corroborated these findings, demonstrating that the proposed AT(N) classifier has prognostic value. AT(N) categories are only modestly influenced by other known factors associated with disease progression. Conclusions: We established CLEIA and ELISA internal cutoffs to discriminate AD patients from amyloid-negative SCD individuals. The results obtained by both methods are not interchangeable but show good agreement. CLEIA is a good and faster alternative to manual ELISA for providing AT(N) classification of our patients. AT(N) categories have an impact on disease progression. AT(N) classifiers increase the certainty of the MCI prognosis, which can be instrumental in managing real-world MCI subjects.

## 1. Introduction

Extracellular amyloid plaques, mainly formed by Aβ1-42 peptide, and intraneuronal neurofibrillary tangles, formed by phosphorylated Tau protein aggregates, are, along with cerebral amyloid angiopathy (CAA) and subsequent neuronal loss, the main neuropathological hallmarks of Alzheimer’s disease (AD) [1,2].

To increase the certainty of AD being the underlying cause of dementia, several AD diagnostic criteria recommend the use of biomarkers tightly associated with AD pathology hallmarks [3,4]. In 2018, the National Institute on Aging and Alzheimer’s Association (NIA-AA) proposed the use of neuroimaging and cerebrospinal fluid (CSF) biomarkers through an AT(N) classification for research purposes [5]. Furthermore, beyond the well-recognized utility for identifying the presence of AD pathology in patients with overt dementia, the biomarker-based prediction of conversion to dementia in people with mild cognitive impairment (MCI) might be relevant for future planning and to identify suitable patients for clinical trials and future disease-modifying treatments [6].

CSF biomarkers have several advantages over imaging techniques, including lower costs, wider availability, short testing time, multi-biomarker analysis, and, in common with neuroimaging biomarkers, longitudinal monitoring [7]. Barriers to using CSF biomarkers are patient reluctance to undergo a lumbar puncture (LP), medical contraindications, and measurement variability due to preanalytical and analytical issues [8]. To overcome this variability, efforts have been made in pre-analytic standardization [9,10], reagent quality control (QC) improvement, the introduction of certified reference materials (CRM), and automatization of CSF analysis. More recently, an international workgroup led by the Alzheimer’s Association developed a simplified and standardized pre-analytical protocol for CSF collection and handling before analysis for routine clinical use [11]. The widespread application of these protocols will help to minimize variability in measurements and to facilitate the implementation of unified cutoff levels across laboratories using the same platform [11]. Despite current progress, to overcome the intrinsic variability of the proposed core AD biomarkers, it is still recommended to establish internal cutoffs for each institution.

Fully automated immunoassay platforms are now available for measuring core AD-biomarkers in CSF [12,13,14]. These platforms have been designed to reduce assay variability, costs, and testing time [15]. Among them, the Lumipulse G600II (Fujirebio Inc., Tokyo, Japan) with the CLEIA based-method is available in our laboratory. In this study, we sought to compare automated Lumipulse G^®^ CLEIA in the Lumipulse platform and manual INNOTEST^®^ ELISA. Moreover, we aimed to establish the cutoffs of CSF biomarkers to differentiate AD patients from amyloid-negative subjective cognitive decline (SCD) individuals measured by two immunoassays. We also applied the same calculated cutoffs to evaluate the agreement between single CSF biomarkers and the Aβ1-42/Aβ1-40 ratio. Most importantly, we evaluated the impact of AT(N) stratification [5] on the progression of a large cohort of MCI subjects.

## 2. Results

### 2.1. Cutoffs of CSF Biomarkers for Both Immunoassays

Patients diagnosed as probable AD (n = 37) and amyloid-negative SCD individuals (n = 45) were included in the cutoff study analysis. Details of the exclusion criteria are reported in Appendix A.

Demographics, *APOE* ε4 status, MMSE scores, and CSF biomarkers (ELISA and CLEIA) are reported in Table 1. AD patients were significantly older, less educated, and had lower MMSE scores than SCD individuals. The AD group had a higher frequency of *APOE* ε4 carriers and comparable gender distribution. For both immunoassays, CSF levels of Aβ1-42, t-Tau, and p181Tau, reported in Appendix A, were significantly different in the AD group compared to the SCD group, whereas Aβ1-40 was not significantly different.

ROC curves obtained for CLEIA and ELISA are shown in Figure 1. The optimal cutoffs are summarized in Table 2. For the CLEIA assay, the cutoffs were 796 pg/mL for Aβ1-42, 412 pg/mL for t-Tau, and 54 pg/mL for p181tau. The highest accuracy was for p181Tau, with an AUC of 0.97, followed by t-Tau with 0.95 and Aβ1-42 with 0.93 (Figure 1A). Aβ1-40 alone had the lowest accuracy, with an AUC of 0.60. Aβ1-42/Aβ1-40 and Tau ratios presented higher accuracy (AUCs of 0.99) when compared with individual biomarkers (Figure 1B). Optimal cutoffs for the ratios were 0.063 for Aβ1-42/Aβ1-40, 1.37 for Aβ1-42/t-Tau, and 11.55 for Aβ1-42 /p181Tau. For the ELISA assay, the cutoffs were 676 pg/mL for Aβ1-42, 367 pg/mL for t-Tau, and 58 pg/mL for p181tau. Aβ1-42 had the highest accuracy, with an AUC of 0.98, followed by t-Tau with 0.95 and p181Tau with 0.92 (Figure 1C). Aβ1-40 alone had the lowest accuracy, with an AUC = 0.55. Aβ1-42/Aβ1-40 had an accuracy of 0.98, similar to Aβ1-42 alone, and a cutoff of 0.069 (Figure 1D). For the other ratios, similarly to that observed with CLEIA, the accuracy was higher when compared with the individual biomarker. Cutoffs were 2.13 for Aβ1-42/t-Tau and 13.73 for Aβ1-42/p181Tau, with an AUC of 0.99 for both.

### 2.2. Comparison of Immunoassays

Comparison by Spearman analysis showed a significant correlation for Aβ1-42 of 0.93 (n = 519), t-Tau of 0.98 (n = 364), p181tau of 0.96 (n = 77), and Aβ1-40 of 0.81 (n = 44), significant at the 0.01 alpha level (Appendix A).

Comparisons by Passing–Bablok regression analyses are shown in Figure 2 and Appendix A. For Aβ1-42 (Figure 2A), t-Tau (Figure 2C), p181Tau (Figure 2E), and Aβ1-40, there was a systematic difference between both methods since the 95% CIs of the intercepts did not include the 0 value: −126.5 (95%CI −159.6 to −92.72) for Aβ1-42, 10.52 (95%CI 1.09 to 20.4) for t-Tau, −28.48 (95%CI −35.33 to −19.63) for p181-Tau, and −2707.21 (95%CI −5338.50 to −233.23) for Aβ1-40. Moreover, for Aβ1-42, t-Tau, and p181Tau, the two methods showed a proportional difference as the 95% CIs of the slopes did not include 1: 1.2 (95%CI 1.15 to 1.26) for Aβ1-42, 1.12 (95%CI 1.09 to 1.14) for t-Tau, and 1.54 (95%CI 1.42 to 1.66) for p181Tau. In contrast, the value for Aβ1-40 was not proportionally different (1.03 [95%CI 0.85 to 1.22]).

Bland–Altman analysis results considering ELISA vs. CLEIA differences are pictured in Figure 2. They showed a mean difference in favor of CLEIA of −1.59% (95%CI −3.01 to −0.17) for Aβ1-42 (Figure 2B), −14.71% (95%CI −15.82 to −13.60) for t-Tau (Figure 2D), and −10.47% (95%CI −16.61 to −4.33) for p181Tau (Figure 2E). The 95%CI of the mean differences did not include zero, indicating that there was a significant systematic difference. For Aβ1-42, mean values above 1200 pg/mL were below the line of equality, indicating that the CLEIA assay gave higher values than ELISA. For p181Tau, the graph suggested a proportional difference in favor of the CLEIA method. The latter observation may be due to the small set of samples analyzed. For all assays evaluated, more than 95% of the measured values were within ±1.96 SD.

### 2.3. Concordance between Immunoassays

To check the utility of the cutoffs described in Table 3 and the concordance of single biomarkers and Aβ1-42/Aβ1-40 for both methods, patients (n = 68) clinically diagnosed with either MCI (n = 48) or dementia (n = 20) were tested. Subsequently, we classified them using AT(N) criteria.

The demographics, *APOE* ε4 status, MMSE scores, and CSF biomarkers (ELISA and CLEIA assessed) of these patients are reported in Appendix A. Briefly, patients with dementia were significantly older and had significantly lower MMSE scores than MCI individuals. In addition, dementia and MCI groups did not show significant differences regarding APOE ε4 allele or gender. With CLEIA, the dementia group had significantly lower Aβ1-42/Aβ1-40 (0.061 ± 0.025 vs. 0.076 ± 0.026, *p* = 0.017), and higher t-Tau (685 ± 481 vs. 418 ± 189, *p* = 0.037) and p181Tau (105 ± 83 vs. 61 ± 34, *p* = 0.018), than the MCI group. For ELISA, the dementia group showed a trend of lower Aβ1-42/Aβ1-40 (0.051 ± 0.021 vs. 0.062 ± 0.022, *p* = 0.053) and higher t-Tau (594 ± 502 vs. 313 ± 145, *p* = 0.028) and p181Tau (81 ± 46 vs. 56 ± 22, *p* = 0.034) than the MCI group. Finally, Aβ1-42 assessed with the CLEIA method (747 ± 321 vs. 875 ± 422, *p* = 0.319) and with the ELISA method (717 ± 337 vs. 816 ± 336, *p* = 0.189) was similar for the dementia and MCI groups.

The agreement between immunoassays was assessed by kappa analysis (Table 3). For p181Tau, t-Tau, and Aβ1-42, the agreement amounted to, respectively, 94% (kappa = 0.882; 95%CI 0.770–0.999), 91% (kappa = 0.824; 95%CI 0.693–0.955), and 90% (kappa = 0.796; 95% CI 0.656–0.936). For Aβ1-42/Aβ1-40, the agreement was 81% (kappa = 0.629; 95%CI 0.461–0.79), i.e., lower than for Aβ1-42 alone. Inspection of the discordant cases obtained with both amyloidosis biomarkers showed seven more A+ cases with CLEIA for Aβ1-42 and 13 more A+ cases with ELISA for Aβ1-42/Aβ1-40. The T(N) profile was identical in both assays, except for one N+ case for CLEIA.

### 2.4. MCI Progression in AT(N) Categories

To further validate the utility of the in-house cutoffs, we evaluated the impact of the proposed stratification on the prognosis of MCI patients. We applied AT(N) classification to 647 MCI subjects with longitudinal data available (mean follow-up 1.75 years). The baseline characteristics of this cohort are depicted in Table 4. Baseline AT(N) patterns were collapsed into four clinically meaningful groups of subjects [5] (Table 5; Figure 3C). Kaplan–Meier survival analysis (time to dementia) demonstrated highly statistically significant differences in the time to conversion among the four groups configured (Figure 3A, *p* = 9.815^−27^). As expected, the most pronounced differences appeared when comparing the extreme AT(N) groups, i.e., normal AT(N) profile versus AD (HR = 7.45 [4.82–11.52]). Interestingly, MCI subjects with amyloidosis (A+T-N-) had a greater risk of progression to dementia than those categorized as SNAP (observed HR = 2.53 [1.50–4.27] vs. HR = 4.58 [2.74–7.67] for the SNAP and brain amyloidosis categories, respectively, compared to the normal ATN group). Observed differences among these intermediate-risk groups reached nominal statistical significance as well (*p* = 0.02).

There are a number of potential covariates that might modify the risk of conversion to dementia in MCI subjects [16,17], including the *APOE* genotype status [18]. For this reason, we decided to investigate potential confounders using Cox proportional-hazard risk models. We analyzed the impact of age, sex, *APOE* genotype, molecular method used (CLEIA vs. ELISA), baseline MMSE, MCI subtype [16], and education as potential confounders. Forward conditional Cox analysis identified that age (*p* = 0.026), baseline MMSE (*p* < 0.001), and MCI subtype (*p* < 0.001) significantly impacted the model. However, after the incorporation of these covariates into the model, the AT(N) classifier remained as the most significant factor associated with MCI progression (adjusted *p*-value = 3.26 × 10^−13^). Adjusted hazard ratios for AT(N) categories ranged from 5.31 [3.34–8.15] for prodromal AD (*p* = 1.97 × 10^−12^) to 2.073 [1.19–3.60] for SNAP (*p* = 0.01). MCI subjects with only brain amyloidosis also converted to dementia 3.7 times more frequently than those subjects with a normal AT(N) profile (adjusted HR=3.69 [2.15–6.33], *p* = 1.99^−6^).

To check relationships among significant factors influencing MCI progression (age, baseline MMSE, MCI subtype), we researched their potential interaction with the proposed AT(N) categories. None of the interaction terms analyzed reached statistical significance (*p* > 0.112). Of note, the Spearman rank coefficient analyses detected a weak but significant correlation among all predictors of MCI progression (Appendix A). We concluded that the AT(N) classifier generated cannot be considered fully orthogonal to other variables associated with progression and is the strongest predictor of conversion to dementia in our real-world clinical series. Having in mind these observations, the estimates derived from a fully adjusted model might be somehow overconservative due to a certain degree of model overfitting.

## 3. Discussion

Here, we reported the established cutoffs of the CSF AD core biomarkers for the manual ELISA and automated CLEIA methods from the ACE CSF cohort. We evaluated the concordance of both immunoassays, identifying a systematic and proportional difference for Aβ1-42, t-Tau, and p181Tau, and only systematic for Aβ1-40. These findings confirmed the need to establish new cutoffs when changing from ELISA to CLEIA in the Lumipulse platform. Utilizing these cutoffs, we applied the AT(N) classification to a subset of participants (MCI and dementia), obtaining good agreement between immunoassays for the single core biomarkers and weaker agreement for Aβ1-42/Aβ1-40. It is known that amyloid forms other than Aβ1-40 and Aβ1-42 can be detected in human CSF. The observed lower agreement of the amyloid based assays might be due to differential affinity to amyloid beta truncated forms [19] or its post translational modifications [20]. We also provided evidence on the clinical utility of AT(N) classifiers in predicting MCI conversion to dementia. Survival analysis of the AT(N) stratified MCI series suggested that this molecular classifier is instrumental in interpreting MCI charts in a real-world setting. However, AT(N) status is not infallible in ruling out the progression of MCI subjects to dementia.

Our study has strengths and limitations. Among the strengths, CSF samples were collected in a single center and using the same protocol, minimizing pre-analytical handling and storage issues affecting the consistency of CSF measurements [21]. Moreover, patients were diagnosed by the same multidisciplinary group of professionals. Importantly, independent evaluation of in-house cutoffs has been conducted in a very large number of MCI patients, being one of the largest single-site MCI longitudinal series reported to date. In fact, our current sample size is larger than that of the multicentric ADNI MCI longitudinal cohort, which is the largest included in the ABIDE study [6]. Hence, we were enough powered to confirm the prognostic value of the proposed AT(N) strata. Notably, we obtained statistically significant results even when intermediate strata where compared (*p* = 0.02). Expanding and longitudinally extending the ACE cohort will be valuable for testing and validating emerging models for the prediction of MCI progression.

Regarding the limitations of this study, on the one hand, the AD patients selected for establishing the CSF cutoffs were diagnosed clinically, meaning that they lacked molecular evidence of AD pathology. Of note, the selection of probable AD patients was performed in a single center, and internal clinical–pathological correlation studies conducted with patients clinically diagnosed with probable AD at ACE who underwent necropsy showed that 95.7% had intermediate or high probability of AD pathology according to the NIA-AA neuropathological guidelines [22] (data not published). On the other hand, even though the SCD subjects selected to establish the CSF cutoffs showed amyloid PET negativity, we cannot rule out that they presented early amyloid deposition and might progress to AD in the future. However, these SCD individuals were participants of the FACEHBI study, with normal cognitive testing, and who had been cognitively stable over the three years after PET evaluation. Furthermore, they had no evidence of cerebral amyloidosis, as measured by two consecutive PET-FBB scans. Thus, none could be classified as being in the preclinical AD phase. We acknowledge that these limitations might potentially impact the exactitude of the cutoffs developed, but we still consider that this is our most affordable solution for selecting cases and controls. Moreover, the MCI progression to dementia analysis demonstrated the clinical value of the proposed cutoffs. Hence, we consider that our strategy could be adopted to address cutoff generation in other memory clinic settings.

Other limitations are the sample size for the cutoff study, which was relatively small (n = 82). The method to select the CSF cutoffs was the maximization of the Youden J index, which balances sensitivity and specificity. This, together with deriving the cutoffs from the population under study, has the risk of overestimating the test’s diagnostic accuracy [14,23,24]. Moreover, the correlation study was less powered for p181Tau and Aβ1-40 than for Aβ1-42 and t-Tau due to the variable number of samples included. In the agreement study, to test the usability of the cutoffs to dichotomize the biomarkers, we assumed that the sample was small and included complex clinical diagnosis (mostly MCI). Moreover, we did not have amyloid PET information available for these patients that could solve the discrepancies between methods and the impact of the Aβ1-40 ratio in classifying the presence of amyloidosis [16].

We observed that cutoff values of Aβ1-42 and t-Tau were higher for CLEIA than for ELISA and similar for p181Tau. As shown in Appendix A, the CLEIA Aβ1-42 cutoff value obtained by us was higher, in agreement with other studies, and the accuracy was better compared with those features provided by the supplier, Fujirebio (AUC = 0.83), or other authors [25,26]. This difference might be due to the selection of controls that they included in their cohorts. Fujirebio and Leitão et al. [25] had controls with other neurological disorders and no amyloid PET. In the study by Bayart et al. [26], controls were selected by clinical diagnosis and (FDG)-PET scan and/or magnetic resonance neuroimaging. In addition, Leitão et al. [25] had only a fraction of AD patients clinically diagnosed with positive amyloid imaging (n = 35). Our pre-analytical protocol was similar to that used by Fujirebio but different from that of Bayart et al. regarding aliquoting tubes.

The CLEIA t-Tau cutoff was very similar to that originally provided by Fujirebio and higher than that published by Bayart et al. and Leitão et al. Our AD diagnostic accuracy features were similar to those of Fujirebio and superior to those of the Bayart et al. study but lower than those of Leitão et al. These differences might be related to subject selection variability and methodological issues. The CLEIA p181Tau cutoff was similar to that of Fujirebio and higher than that of Leitão et al. No comparison was performed with the CLEIA cutoffs reported by Alcolea et al. [14] from the Sant Pau Initiative of Neurodegeneration (SPIN) cohort, since these were determined by optimizing their agreement with 18F-Florbetapir PET amyloid imaging results instead of clinical diagnosis. Moreover, the SPIN cohort was composed of heterogeneous presentations of neurological disorders [27]. Despite the methodological differences, our Aβ1-42/Aβ1-40 cutoff (0.063) was almost identical to that of Sant Pau (0.062) and slightly lower than that reported by Leitão et al. [25] and Fujirebio (0.068 and 0.069). This agreement between independent cutoffs reinforces the reproducibility of the CLEIA method.

Linear correlation between immunoassays was very high for Aβ1-42, t-Tau, and p181Tau and moderate for Aβ1-40, as shown by others [14,25,26]. Both a significant systematic and a proportional difference were observed between methods for all biomarkers except for Aβ1-40. Furthermore, a deviation from linearity was observed for ELISA values of Aβ1-42 and t-Tau over 1200 pg/mL and 1000 pg/mL, respectively. These concentration values are around the upper limits of the ELISA assay ranges for both biomarkers. Similar behavior was shown elsewhere only for Aβ1-42(32). This must be taken into consideration when performing the comparison of historical ELISA data with newly measured cases with Lumipulse. Bland–Altman analysis for Aβ1-42, t-Tau, and p181Tau showed an average difference between methods in favor of the CLEIA assay. More samples should be tested for p181Tau because the Bland–Altman graph suggested that this may be a proportional difference. In summary, we showed that although the immunoassays used similar antibody combinations, measurements were not interchangeable, and both values cannot be mixed in clinical studies. Therefore, datasets analyzed with the two methods should be analyzed separately, standardized, and then meta-analyzed.

Confirming previous reports [27], both CLEIA and ELISA Aβ1-40 showed a lack of diagnostic value when used as a unique biomarker. However, Aβ1-40 improved the agreement with amyloid PET visual status [14,28,29] and can be instrumental in the normalization of Aβ1-42, introducing the inter-individual variation in the total amyloid load [30]. For CLEIA, Aβ1-42/Aβ1-40 was better in distinguishing AD from amyloid-negative SCD, whereas for ELISA, Aβ1-42/Aβ1-40 and Aβ1-42 showed similar accuracy (Table 2). The other ratios (Aβ1-42/t-Tau and Aβ1-42/p181Tau) for ELISA showed an improvement when another biomarker was included (p181Tau and t-Tau).

We evaluated whether our cutoffs for both methods could classify single biomarkers and Aβ1-42/Aβ1-40 similarly. Assay concordance was good for the core biomarkers, whereas Aβ1-42/Aβ1-40 agreement was lower than for Aβ1-42 alone (Table 4). As mentioned, only with the CLEIA immunoassay did Aβ1-42/Aβ1-40 show improved capacity to distinguish AD from amyloid-negative SCD when establishing the cutoff. Similarly, with CLEIA Aβ1-42/Aβ1-40, we observed a significant difference between MCI and AD, whereas with the ELISA ratio, we only observed a trend. These suggest that the decrease in agreement between Aβ1-42/Aβ1-40 measured by the two methods may be related to the ELISA performance.

As previously observed [31], CSF biomarkers cannot predict progression to AD or dementia perfectly. Covariates modulating prediction might vary from one series to another due to ascertainment differences among cohorts. Other demographic factors, clinical, and even ethnic factors might impact the personal prognosis as well [30]. For all these reasons, routine biomarker interpretation is sometimes difficult for clinicians, especially in patients with MCI [31]. Despite these caveats, universal models for providing a personal risk estimation are under development [6]. Generalization of current prediction models will require further research. We are progressively increasing the ACE CSF cohort and we feel that this effort might be instrumental in the independent testing and validation of current and future models for evaluating the individual risk of conversion to dementia in MCI cases.

In our study, a proportion (12.6%) of MCI subjects with normal CSF profile progressed to dementia within a mean follow-up of 1.75 years. Despite their normal baseline AT(N) status, all of them (100%) were endorsed as possible or probable AD by our clinicians in longitudinal evaluations. Only follow-up CSF analysis or postmortem studies could help to clarify whether these cases are true phenocopies (without AD pathology), misdiagnoses with alternative pathological changes (e.g., LATE), or genuine AD patients with very incipient pathology (below established CSF cutoffs) at the time of the lumbar puncture. To further research this issue, we plan to start offering follow-up CSF evaluations to MCI patients with overt cognition decline irrespective of their baseline AT(N) status. Alternatively, the development of earlier biomarkers, appearing in advance of core AT(N) positivity, might be also instrumental in identifying future progressors even with normal AT(N) profiles.

Conversely, a large proportion MCI subjects classified as prodromal AD (57%) according to their ATN profile converted to dementia within the follow-up observational window (mean follow-up of 1.75 years). Survival analyses suggested that nearly 90% of MCI individuals with a CSF-based diagnosis of prodromal AD would progress to dementia in less than four years (Figure 3). Notably, MCI subjects with overt amyloidosis (A+T-N-) showed a worse prognosis than those classified as SNAP. Nevertheless, MCI individuals with these mixed AT(N) patterns (overt amyloidosis and SNAP) might be communicated as having an intermediate risk of progression to dementia. We feel that a larger observational period will increase the accuracy of the prognosis associated with AT(N) categories.

With this information, we could advise about the existence of a good (normal profile), intermediate (brain amyloidosis and SNAP), or poor (prodromal AD) prognosis of MCI cases according to their CSF ATN profile. However, we should be very cautious in explaining the results, indicating that a negative test is non-definitive and does not equal a permanent absence of risk of progression.

In conclusion, in this study, we described the clinical CSF cutoff values to discriminate AD patients from amyloid-negative SCD individuals for standard ELISA and automated CLEIA at the ACE’s Research Center. In addition, we showed that automated CLEIA using the Lumipulse platform results and those obtained with manual ELISA were not interchangeable but showed good agreement. We propose automated CLEIA as a good alternative to standard ELISA together with the use of Aβ1-42/Aβ1-40, which contributes to increasing the accuracy of amyloid determination. AT(N) categories have a strong impact on statistical models for predicting disease progression in MCI individuals. In the routine practice of memory clinics, the AT(N) classifiers can improve the certainty of the prognosis predictions requested by MCI subjects and their families.

## 4. Methods

### 4.1. Participants

The ACE Alzheimer Center Barcelona (ACE) CSF program started in 2016 as an essential research activity of the IMI2-ADAPTED project. This project aimed to disentangle the molecular mechanisms of *APOE*-associated risk alleles to AD (https://www.imi-adapted.eu/ accessed on 11 May 2022). To this end, we initiated the generation of a long-term repository of biomaterials that can be instrumental for deciphering the molecular basis of AD and related disorders.

Beyond our research objectives, from a strictly clinical point of view, we also had the objective of empowering of our clinicians with molecular tools for their daily activities. Consequently, we started offering voluntary (and informed consented) LP to (a) individuals with MCI and dementia evaluated at the Memory Clinic of ACE (Barcelona, Spain) [32]; (b) participants of the ACE Healthy Brain Initiative (FACEHBI) [33], a long-term observational study for identifying biomarkers of preclinical AD in healthy individuals with SCD; and (c) participants of the BIOFACE study, a Prospective Study of Risk Factors, Cognition, and Biomarkers in a Cohort of Individuals with Early-Onset Mild Cognitive Impairment [34].

All participants completed neurological, neuropsychological, and social evaluations. A consensus diagnosis was assigned to each patient by a multidisciplinary team of neurologists, neuropsychologists, and social workers [32]. All subjects were examined with the Spanish version of the Mini-Mental State Examination (MMSE) [35,36], the memory part of the Spanish version of the 7 Minute test [37], the Spanish version of the Neuropsychiatric Inventory Questionnaire (NPI-Q) [38], the Hachinski Ischemia Scale [39], the Blessed Dementia Scale [40], and the Clinical Dementia Rating (CDR) scale [41], and a comprehensive neuropsychological battery of ACE (N-BACE) [42]. Neuroimaging with magnetic resonance imaging (MRI) or computerized tomography (CT) scans were available.

Dementia was defined according to the DSM-V criteria [43]. SCD [44] refers to the perception of memory or other cognitive problems without any evidence of impairment using standardized cognitive tests. Those SCD individuals from the FACEHBI cohort included in the study were cognitively stable for 3 years, with no evidence of brain amyloidosis in 18F-Florbetaben-labeled positron emission tomography (FBB-PET).

The underlying etiologies of the cognitive deficits within the dementia group were classified according to the following criteria: the 2011 NIA-AA for Alzheimer’s disease [3,5], the National Institute of Neurological Disorder and Stroke and Association Internationale pour la Recherche et l’Enseignement in Neurosciences criteria (NINDS-AIREN) for vascular dementia (VD) [45], Frontotemporal Dementia (FTD) [46], and for Lewy body dementia (LBD) [47].

#### MCI Longitudinal Cohort

CSF results with associated follow-up data were available from 647 MCI patients. MCI was defined using Petersen’s criteria [48,49,50]. MCI patients were evaluated and followed up at a single site (ACE). Baseline and follow-up data were obtained between 2016 and 2021. All MCI participants were assessed as previously reported [50,51]. Of note, CSF biomarker results were not included as part of the clinical diagnosis procedure of the institution and were not used for initial diagnostic endorsement in the memory clinic. Follow-up assessments were conducted for MCI individuals on an approximately annual basis. The MMSE and NBACE battery were measured in all visits. Dementia conversion was defined using previously published criteria [51]. *APOE* genotyping was performed using genomic DNA obtained from whole blood collected in BD Vacutainer tubes (K2-EDTA). DNA extraction was performed using DNA Chemagen technology (Perkin Elmer, Waltham, Massachusetts, USA). Genotypes were determined by TaqMan probe analysis in the Real-Time PCR QuantStudio3 (ThermoFisher, Waltham, Massachusetts, USA) system or extracted from Affymetrix Axiom SP biobank arrays processed as previously described [51,52].

### 4.2. CSF Sampling and Analysis

An LP procedure was proposed to patients with MCI and dementia evaluated at the ACE Memory Clinic. For SCD individuals, LP was proposed in the context of the FACEHBI study [33]. The collection protocol followed consensus recommendations [9] (Appendix A). CSF was collected passively in 10 mL polypropylene tubes (Sarstedt Ref 62.610.018) and centrifuged (2000× *g* 10 min at 4 °C) within 2 h of acquisition. After centrifugation, the fluid was aliquoted into polypropylene tubes (Sarstedt Ref 72.694.007) and stored at −80 °C until analysis.

Concurrently matched samples of saliva, serum, plasma, and cell pellets were obtained from each subject for future investigations and are part of the ACE collection registered in ISCIII with the code C.0000299.

On the day of the analysis, two aliquots per patient were thawed at room temperature and tubes were vortexed for 5–10 s. The four biomarkers were quantified directly from the storage tube with the Lumipulse G600II automated platform (Fujirebio Europe, Göteborg, Sweden). We used an adaptor to fit the tubes in the equipment. The other aliquot was used to test the biomarkers using a standard ELISA immunoassay (INNOTEST^®^, Fujirebio Europe, Göteborg, Sweden). Reagents used were from different batches. In parallel to the regular CSF analyses, ACE participated in the AA external QC program for CSF biomarkers [52].

For Lumipulse^®^ assays, in accordance with the supplier specifications, each biomarker was run in parallel with a 3-point calibration curve (per duplicate) and three controls included in the kit (low, medium, and high concentration). The mean inter-assay coefficients of variation for the Lumipulse controls (low, medium, and high concentration) were 2.8%, 3.2%, and 3.0% for Aβ1-42 (n = 21); 6.3%, 3.7%, and 5.1% for t-Tau (n = 15); 2.9%, 0.8%, and 2.3% for p181Tau (n = 5), and 2.5%, 1.2%, and 3.3% for Aβ1-40 (n = 5), respectively. For ELISA INNOTEST^®^ assays, a 6-point calibration curve (per duplicate) and 2 controls (low and high concentration) were included on each plate. For both assays, QC were within the target ranges specified by the supplier.

The results of the Lumipulse G Aβ1-42 presented in this study were standardized using CRMs^40^. The aim was to harmonize immunoassays of Aβ1-42 for the results to be comparable across different platforms. Briefly, values of the calibration standards of the Lumipulse G600II were adapted to the CRM, resulting in an adjustment of concentrations that was linearly proportional throughout the entire measurement range.

### 4.3. Ethical Considerations

The LP consent was approved by the ethical committee of the Hospital Clinic i Provincial de Barcelona (Barcelona, Spain) in accordance with Spanish biomedical laws (Law 14/2007, 3 July, regarding biomedical research; Royal Decree 1716/2011, 18 November) and followed the recommendations of the Declaration of Helsinki. For more details, see the Appendix A.

#### 4.3.1. Study 1: Determination of Cutoff for Both Immunoassays

For the determination of the cutoff, AD patients and SCD individuals were selected as cases and controls, respectively, according to the criteria shown in Table 5.

#### 4.3.2. Study 2: Comparison of Immunoassays

CSF samples from consecutive patients evaluated at the Memory Clinic were tested on the Lumipulse and on the Innotest assay. Aβ1-42 (n = 527), t-Tau (n = 399), Aβ1-40 (n = 44), and p181Tau (n = 77) were evaluated. The difference in sample size was due to the different launching dates of the Lumipulse assays. Details of them and descriptive statistics for correlation studies are available in Appendix A.

Before comparison, 1% of cases were excluded for Aβ1-42 and t-Tau assessed by CLEIA assay. Likewise, 5% and 3.6%, respectively, were excluded from the ELISA assay. These values were beyond the pre-specified assay range and the samples were diluted if recommended by the provider. No values were excluded for p181Tau and Aβ1-40 assessed by CLEIA and ELISA assays.

#### 4.3.3. Study 3: Concordance between Immunoassays

To evaluate the agreement between the CLEIA and ELISA immunoassays of the three single biomarkers and Aβ1-42/Aβ1-40, the four biomarkers were tested in an independent subset of patients clinically diagnosed either with MCI or dementia (n = 68). Each biomarker’s results were dichotomized using the corresponding cutoffs established here. The AT(N) scheme [5] was constructed taking into account that “A” is accepted as an indicator of Aβ deposition for both CSF Aβ1-42 and Aβ1-42/Aβ1-40^41^. “T” and “N” correspond to p181Tau and to t-Tau, respectively.

#### 4.3.4. Study 4: MCI Progression

To evaluate the prognostic value of the AT(N) classifier [5], an identical AT(N) scheme as defined in study 3 was applied to 647 MCI subjects with lumbar puncture and with follow-up data available. In order to facilitate interpretation, the eight AT(N) categories were collapsed into four different groups of patients, as previously suggested [5]: (1) subjects with a normal biomarker profile (“Normal profile”, A-T-N-, n = 190, 29.3%); (2) subjects with detectable brain amyloidosis but without evidence of tau or phospho-tau burden (“Brain amyloidosis or Alzheimer’s pathologic change”, A+T-N-, n = 88, 13.6%); (3) suspected non-amyloid pathology (SNAP = A-T+N-/A-T-N+/A-T+N+, n = 125, 19.3%) and prodromal AD (A+T+N+/A+T-N+/A+T+N-, n = 244, 37.7%).

### 4.4. Statistical Analysis

For statistical analysis, we used SPSS (version 26.0 for Windows, IBM, Armonk, New York, USA). Differences among groups were assessed using the Chi-Square test for categorical variables and *t*-test for quantitative variables. The distribution of biomarkers was checked for normality with the Shapiro–Wilk test. Due to non-normality of the distribution of biomarkers, the Mann–Whitney U test was used to compare biomarker levels among the diagnostic groups.

Receiver operating characteristic (ROC) curve and area under the curve (AUCs) were calculated and plotted for every biomarker (Aβ1-42, Aβ1-40, t-Tau, and p181Tau) and ratio (Aβ1-42/Aβ1-40, Aβ1-42/t-Tau, and Aβ1-42/p181Tau). Cutoffs for each single biomarker and the ratios were selected from the ROC curve coordinates according the highest Youden J index (Sensitivity + Specificity − 1).

To study the magnitude of the association between both methods, the Spearman’s Rho correlation coefficient was calculated. Determination of the proportional difference (slope) and constant difference (intercept) was performed using Passing–Bablok regression (R software, R Foundation, Vienna, Austria). Comparison of both methods was performed using Bland–Altman analysis to assess the agreement between the two platforms considering ELISA vs. CLEIA differences (expressed in percentages). Concordance between analytical platforms of the core AD biomarkers and Aβ1-42/Aβ1-40 was performed using kappa agreement.

The impact of the AT(N) stratification on the risk of conversion to dementia was assessed using Kaplan–Meier curves. For this purpose, MCI subjects were divided into four AT(N) categories. The AT(N) stratification was based on the same empirical cutoffs previously calculated. To research potential confounders of AT(N) categories, Cox proportional-hazard regressions adjusted for multiple covariates were conducted (age at lumbar puncture, sex, MMSE score at baseline, formal education (years), MCI clinical categories (amnestic/non-amnestic>probable/possible) [50], or biomarker determination methods (ELISA or CLEIA)). We used the normal profile category (A-T-N-) as the reference category for comparisons. The significance level was set at alpha = 0.05.

## Figures and Tables

**Figure 1 ijms-23-06891-f001:**
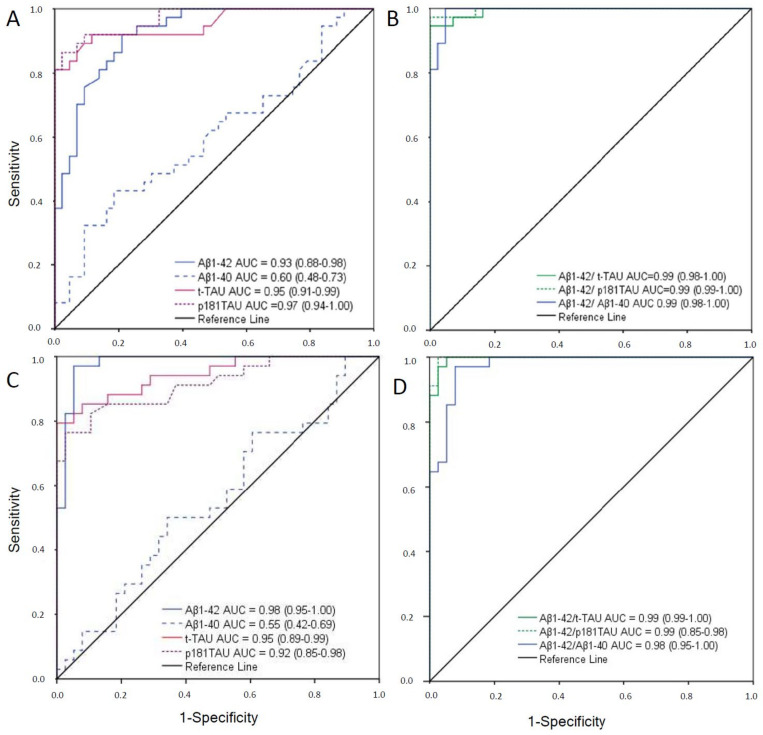
ROC analysis compared to clinical diagnosis as specified in Study 1. (**A**). For each CSF biomarker measured with CLEIA. (**B**). For each CSF biomarker ratio measured with CLEIA. (**C**). For each CSF biomarker measured with ELISA. (**D**). For each CSF biomarker ratio measured with ELISA. Abbreviations: AUC: area under the curve; CLEIA, CSF, ELISA, ROC.

**Figure 2 ijms-23-06891-f002:**
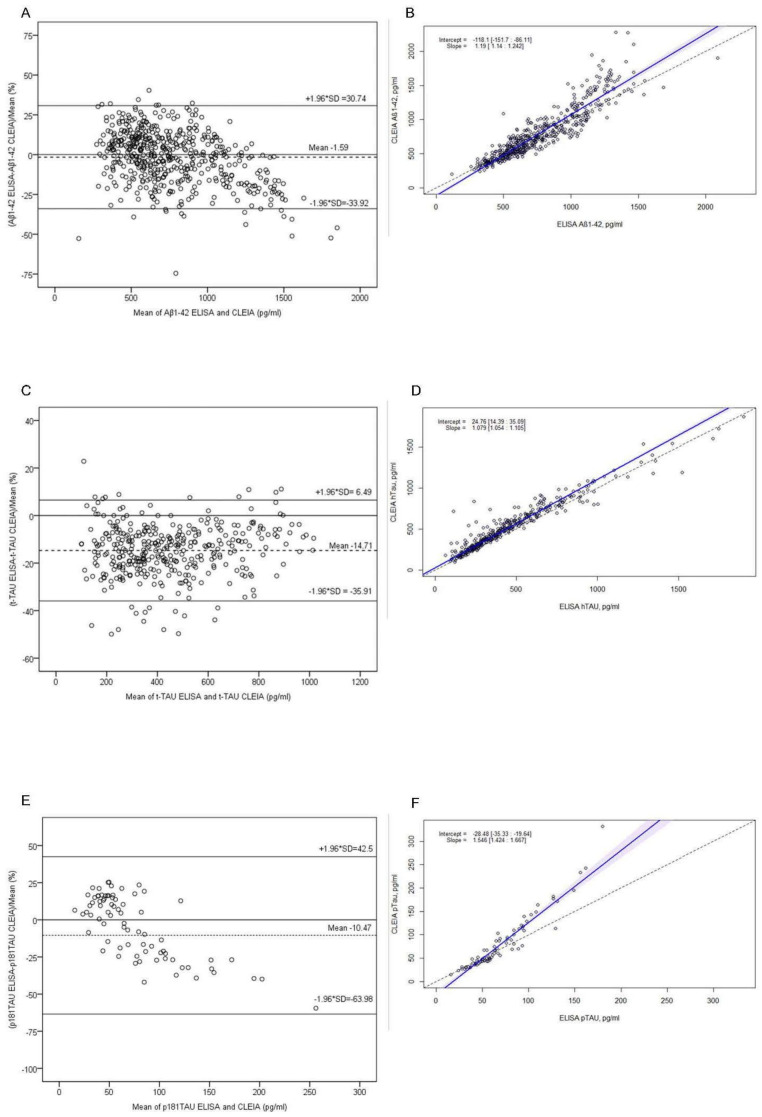
Bland–Altman (**A**,**C**,**E**) and Passing–Bablok analysis (**B**,**D**,**F**) for Aβ42, hTAU, and p181TAU biomarkers comparing CLEIA with ELISA assay for Aβ1-42 (n = 519), for t-Tau (n = 399), and for p181Tau (n = 77). For Bland–Altman analysis, the solid line shows 0 or no difference, while the dotted line shows the mean difference ± 1.96 standard deviation (SD). For Passing–Bablok analysis, dotted lines represent the equation x = y (identity line) and the blue areas show the 95% CI of the regression lines. Abbreviations: CLEIA, ELISA.

**Figure 3 ijms-23-06891-f003:**
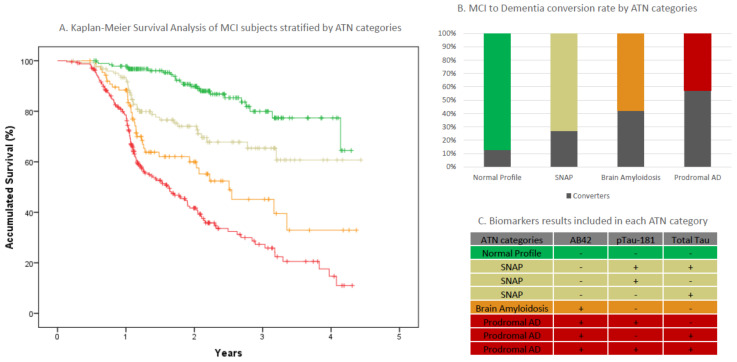
MCI to dementia progression analysis in ACE Alzheimer Center Barcelona CSF cohort. (Panel (**A**)). Kaplan-Meier survival analysis. (Panel (**B**)). Conversion rate observed in AT(N) categories. (Panel (**C**)). AT(N) categories included in each stratum. Abbreviations: MCI, CSF.

**Table 1 ijms-23-06891-t001:** Cutoff study: demographic, clinical, genetic, and biomarker data.

	SCD	AD	Chi/T/UTest	*p*-Value
n (%)	45 (55%)	37 (45%)		
Age, years	65.6 ± 5.7	74.3 ± 8.4	5.35	<0.0001
Gender, Female/Male (% Female)	26/19 (58%)	29/8 (78%)	3.03	0.082
APOEε4 ^a^ +/− (%+)	7/38 (16%)	16/21 (43%)	6.40	0.011
MMSE ^b^ score	29.6 ± 0.7	20.7 ± 4.9	11.04	<0.0001
Education, years (n = 79)	13.0 ± 3.3	7.7 ± 4.0	6.41	<0.0001
**CSF biomarkers, ELISA**				
Aβ1-42, pg/mL	1053 + 261	527 ± 112	36.00	<0.0001
Aβ1-40, pg/mL (n = 72)	11567 ± 3462	12216 ± 3566	578.00	0.443
t-Tau, pg/mL	237 ± 79	726 ± 421	92.00	<0.0001
P181Tau, pg/mL	44 ± 12	100 ± 49	137.00	<0.0001
Aβ1-42/Aβ1-40 (n = 72)	0.096 ± 0.023	0.045 ± 0.012	32.00	<0.0001
Aβ1-42/t-Tau	4.93 ± 1.56	0.93 ± 0.49	5.00	<0.0001
Aβ1-42/p181Tau	25.50 ± 6.87	6.23 ± 2.5	4.00	<0.0001
t-Tau/Aβ1-42	0.042 ± 0.015	0.1913 ± 0.091	5.00	<0.0001
P181Tau/Aβ1-42	0.2248 ± 0.098	1.3938 ± 0.778	4.00	<0.0001
**CSF biomarkers, CLEIA**				
Aβ1-42, pg/mL	1171 ± 395	568 ± 179	121.00	<0.0001
Aβ1-40, pg/mL (n = 80)	13093 ± 3654	14576 ± 4207	633.00	0.117
t-Tau, pg/mL	300 ± 91	825 ± 431	80.50	<0.0001
P181Tau, pg/mL (n = 80)	40 ± 11	145 ± 87	48.00	<0.0001
Aβ1-42/Aβ1-40 (n = 80)	0.088 ± 0.014	0.039 ± 0.008	11.00	<0.0001
Aβ1-42/t-Tau (n = 82)	4.13 ± 1.36	0.81 ± 0.42	10.00	<0.0001
Aβ1-42/p181Tau (n = 80)	29.6 ± 8.1	4.9 ± 3.0	6.00	<0.0001
t-Tau/Aβ1-42 (n = 82)	0.27 ± 0.12	1.48 ± 0.6	10.00	<0.0001
P181Tau/Aβ1-42 (n = 80)	0.04 ± 0.01	0.25 ± 0.13	6.00	<0.0001

^a^ APOEε4: apolipoprotein allele e4 carriers, ^b^ MMSE: Mini-Mental State Examination. Data are presented as mean (SD) unless otherwise specified. *p*-values were calculated by comparing SCD individuals and AD patients using Student *t*-test, χ2, and, for biomarkers, Mann–Whitney U test.

**Table 2 ijms-23-06891-t002:** ROC ^a^ analysis of CSF ^b^ biomarkers for distinguishing AD from amyloid-negative SCD.

Immunoassay	AUC ^c^ (95%IC)	Cutoff ^d^	Youden J Index	Sensitivity	Specificity
**Innotest ELISA** ^e^					
Aβ1-42	0.98 (0.95–1.00)	<676	0.90	95	96
Aβ1-40 (n = 72)	0.55 (0.42–0.69)	<10,530	0.16	77	40
t-Tau	0.95 (0.89–0.99)	>367	0.80	87	93
p181Tau	0.92 (0.85–0.98)	>58	0.72	81	91
Aβ1-42/Aβ1-40(n = 72)	0.98 (0.95–1.00)	<0.069	0.89	97	92
Aβ1-42/t-Tau	0.99 (0.99–1.00)	<2.13	0.95	97	98
Aβ1-42/p181Tau	0.99 (0.99–1.00)	<13.73	0.98	100	98
**Lumipulse CLEIA** ^f^					
Aβ1-42	0.93 (0.88–0.98)	<796	0.72	92	80
Aβ1-40 (n = 80)	0.60 (0.48–0.73)	<15,158	0.18	49	60
t-Tau	0.95 (0.91–0.99)	>412	0.81	92	89
p181Tau (n = 80)	0.97 (0.94–1.00)	>54	0.83	92	91
Aβ1-42/Aβ1-40 (n = 80)	0.99 (0.98–1.00)	<0.063	0.95	100	95
Aβ1-42/t-Tau	0.99 (0.98–1.00)	<1.37	0.95	95	100
Aβ1-42/p181Tau	0.99 (0.99–1.00)	<11.55	0.97	97	100

^a^ ROC: receiver operating characteristic, ^b^ CSF: cerebrospinal fluid, ^c^ AUC: area under the curve, ^d^ cutoffs for single biomarkers are given in pg/mL, ^e^ ELISA: enzyme-linked immunosorbent assay, ^f^ CLEIA: chemoluminescence enzyme immunoassay.

**Table 3 ijms-23-06891-t003:** Agreement of single biomarkers and Aβ42/Aβ40 ratio measured by ELISA and CLEIA immunoassays.

ELISA	CLEIA		Kappa	CI 95%	Agreement (%)
	Negative	Positive			
Aβ42					
Negative	32	7	0.796	0.656–0.936	90
Positive	0	29			
t-Tau					
Negative	33	6	0.824	0.693–0.955	91
Positive	0	29			
p181Tau					
Negative	33	2	0.882	0.770–0.999	94
Positive	2	31			
Aβ42/Aβ40					
Negative	25	0	0.629	0.461–0.79	81
Positive	13	30			

Data are expressed as the number of patients.

**Table 4 ijms-23-06891-t004:** MCI longitudinal study: demographics, clinical, genetic, and neuropsychological data.

	All MCIs	A-T-N- Normal	A+T-N- Amyloidosis	A-(TN)+ SNAPS	A+(TN)+ Prodromal AD
n (%)	647	190 (29.4)	88 (13.6)	125 (19.3)	244 (37.7)
Age, years (sd)	72.8 (7.78)	69.3 (9)	72.4(7.6)	73.8(7.1)	75.1(6.1)
Sex, (n, % Female)	347 (53.6)	98 (51.6)	42 (47.7)	68 (54.4)	129 (57)
APOEε4 carriers (%+) *	32.7%	12.7%	33%	26.8%	53%
Mean baseline MMSE score (sd)	25.55 (3.2)	26.3 (3.1)	25.5 (3)	25.8 (3.4)	24.9 (3.3)
Education mean years (s)	8.1 (4.8)	8.3 (4.2)	8 (6.9)	7.8 (4.6)	8.1 (4.3)
Follow-up time mean years (sd)	1.75 (0.9)	2.11 (0.9)	1.64 (0.9)	1.86 (0.9)	1.44 (0.9)
Dementia conversion rate n (%)	234 (36.2)	24 (12.6)	37 (42)	34 (27.2)	139 (57)
Non-AD conversions n (%) **	39 (16.7)	15 (62.5)	9 (24.3)	11 (32.4)	4 (2.9)
ELISA/CLEIA/na (n)	346/293/8	114/70/6	44/44/0	74/50/1	114/129/1
NPS clinical categories (n) ***	107/25/283/223/9	49/7/98/33/3	15/3/42/27/1	25/3/52/42/3	18/12/91/121/2

Note: SNAP or A-(TN)+ stratum includes three ATN categories (A-T+N+, A-T+N-, and A-T-N+). Same for prodromal AD or A+(TN)+ stratum, which includes A+T+N+, A+T+N-, and A+T-N+ categories. * APOE genotype available only for individuals consenting to genetic studies and with DNA available (n = 464). ** Non-AD dementia conversion is declared when dementia etiology endorsed by the neurologist is not Alzheimer’s disease. *** NPS: MCI neuropsychological categories according to Espinosa et al. 2013. Possible non-amnestic/probable non-amnestic/possible amnestic/probable amnestic/not available.

**Table 5 ijms-23-06891-t005:** Cutoff study: inclusion and exclusion criteria for cases and controls.

Cases
Inclusion criteria
- Clinically diagnosed as probable AD ^a^ according to the McKhann criteria (3) - Mild or moderate stages equivalent to GDS ^b^ score 4–5
Exclusion criteria
- Multiple or extensive infarcts or severe white matter hyper-intensity burden in the neuroimaging study - Core features of LBD ^c^ - Prominent features of behavioral variant FTD ^d^, semantic variant primary progressive aphasia or non-fluent/agrammatic variant primary progressive aphasia - Other concurrent, active neurological disease - Evidence of a non-neurological comorbidity or use of medication that could have a substantial effect on cognition
Controls
Inclusion criteria
- Individuals with SCD ^e^, participants of the FACEHBI ^f^ cohort (mean age, 65.8 ± 7.1 years; 62.5% women), and screened for brain amyloidosis with FBB-PET ^g^ performed as described (14) - Underwent an LP - No evidence of brain amyloidosis in an FBB-PET with a SUVR ^h^ cutpoint < 1.45, at visit 2 aligned in time with the LP (screening two years later)
Exclusion criteria
- Cognitive worsening leading to a diagnosis of ^i^ MCI in any of the FACEHBI study visits before the LP - And/or positivity of brain amyloidosis in an FBB-PET with SUVR cutpoint > 1.45, at visit 2

^a^ AD: Alzheimer’s disease, ^b^ GDS: Global Deteriorating Scale, ^c^ LBD: Lewy Body Dementia, ^d^ FTD: Frontotemporal Dementia, ^e^ SCD: subjective cognitive decline, ^f^ FACEHBI: Fundació ACE Health Brain Initiative, ^g^ FBB-FET: ^18^F-Florbetaben-labeled positron emission tomography, ^h^ SUVR: global standardized uptake value ratio, ^i^ MCI: mild cognitive impairment.

## Data Availability

The datasets used and/or analyzed during the current study is available from the corresponding author on reasonable request.

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
