# Peer review of "Establishing In-House Cutoffs of CSF Alzheimer’s Disease Biomarkers for the AT(N) Stratification of the Alzheimer Center Barcelona Cohort"

_ijms, 2022, doi:10.3390/ijms23136891_

Round 1

Reviewer 1 Report

In this study Orellana and colleagues compared standard ELISA and automated CLEIA and calculated the clinical CSF cutoff values able to discriminate AD patients from amyloid-negative individuals with subjective memory complains. They showed that automated CLEIA using the Lumipulse platform results and those obtained with manual ELISA were not interchangeable but showed good agreement. Considering single markers, they obtained good agreement between immunoassays, especially for t-tau and p-tau; a weaker agreement was obtained for Aβ1-42/Aβ1-40. Finally, they confirmed the clinical utility of AT(N) classifiers in predicting MCI conversion to dementia. This is a very usefull and well done study, with a great impact on clinical practice.

Minor points:

1-      It is known that other forms other than abeta-140 and abeta 1-42 can be detected in human CSF: authors should discuss whether the low agreement of the two assays might be due to different abeta isoforms detected

2-      Authors declare that their study is “ well powered to test the prognostic value of the proposed AT(N) strata and to validate future predictive models to be applied to MCI patients” please provide power calculation.

3-      Table 2: please remove the column “All” which may be confusing because of statistical analysis

Author Response

Thanks to reviewer 1 for his/her positive comments. 

Minor points:

1-      It is known that other forms other than abeta-140 and abeta 1-42 can be detected in human CSF: authors should discuss whether the low agreement of the two assays might be due to different abeta isoforms detected

Answer: Thanks for this commentary. In accordance with the reviewer's suggestion, we added this possibility in the discussion section

2-      Authors declare that their study is “ well powered to test the prognostic value of the proposed AT(N) strata and to validate future predictive models to be applied to MCI patients” please provide power calculation.

Answer: thanks for this commentary. We put this sentence because, compared with currently available datasets, we have one of the biggest single-site datasets for testing predictive models for MCI to AD phenoconversion reported to date (see ABIDE sample size datasets used in references 6,12 and 49). Indeed, we detected statistically significant differences among ATN categories using Cox-proportional Hazard risk models adjusted by key covariates.  It is well established that when a sample is selected, outcomes are no longer random and post hoc power analysis becomes meaningless for this study sample. (see https://www.vims.edu/people/hoenig_jm/pubs/hoenig2.pdf )

To avoid any confusion, we have slightly modified the sentence in the manuscript.

3-      Table 2: please remove the column “All” which may be confusing because of statistical analysis

Answer: Thanks for this idea. In accordance with the reviewer's suggestion, we have removed "All" column in table 2

Reviewer 2 Report

Orellana A. et. al. Investigated the variability between Fujirebio’s ELISA and Lumipulse assays in core AD biomarker measurements and discussed the importance of the use of in-house cut-off values for these markers in assessing AT(N) classification of dementia cases in clinic. Authors clearly explained the difference between two assays using appropriate statistics and experimental designs. Overall, this is an intriguing study and the manuscript is very well written. I would recommend acceptance of this manuscript.

Author Response

Orellana A. et. al. Investigated the variability between Fujirebio’s ELISA and Lumipulse assays in core AD biomarker measurements and discussed the importance of the use of in-house cut-off values for these markers in assessing AT(N) classification of dementia cases in clinic. Authors clearly explained the difference between two assays using appropriate statistics and experimental designs. Overall, this is an intriguing study and the manuscript is very well written. I would recommend acceptance of this manuscript.

Answer: thanks to reviewer 2 for her/his kind commentary. 

Round 2

Reviewer 1 Report

I have no further concerns / suggestions